# Impact Energy Absorption Analysis of Shape Memory Hybrid Composites

**Huma Ozair** [1,*], **Muhammad Atiq Ur Rehman** [1], **Abrar H. Baluch** [1], **Khurram Yaqoob** [2], **Ibrahim Qazi** [1] **and Abdul Wadood** [1,*]

1   Department of Materials Science and Engineering, Institute of Space Technology (IST), Islamabad 44000, Pakistan
2   School of Chemical and Materials Engineering, National University of Science and Technology (NUST), Islamabad 44000, Pakistan
*   Correspondence: huma.ozair@gmail.com (H.O.); abdul.wadood@mail.ist.edu.pk or wadood91@gmail.com (A.W.)

**Abstract:** Shape memory hybrid composites are hybrid structures with fiber-reinforced-polymer matrix materials. Shape memory wires due to shape memory/super-elastic properties exhibit a pseudo-elastic response with good damping/energy absorption capability. It is expected that the addition of shape memory wires in the glass-fiber-reinforced-polymer matrix composite (GFRP) will improve their mechanical and impact resistant properties. Stainless-steel wires are also expected to improve the impact resistance properties of GFRPs. In this research work, we investigated the effect of addition of shape memory wires and stainless-steel wires on the impact resistance properties of the GFRP and compared our results with conventional GFRPs. Super-elastic shape memory alloy wires and stainless-steel wires were fabricated as meshes and composites were fabricated by the hand-layup process followed by vacuum bagging and the compression molding setup. The shape-memory-alloy-wires-reinforced GFRP showed maximum impact strength followed by stainless-steel-wires-reinforced GFRPs and then conventional GFRPs. The effect of the energy absorption capability of super-elastic NiTi wires owing to their energy hysteresis was attributed to stress-induced martensitic transformation in the isothermal regime above the austenite transformation temperature. The smart shape memory wires and stainless-steel-wires-based hybrid composites were found to improve the impact strength by 13% and 4%, respectively, as compared to the unreinforced GFRPs. The shape-memory-reinforced hybrid composite also dominated in specific strength as compared to stainless-steel-wires-reinforced GFRPs and conventional GFRPs.

**Keywords:** shape memory wires; martensitic transformation; shape memory hybrid composites; impact energy absorption

## 1. Introduction

Functional materials have the capability to act smartly in structures providing properties that cannot be directly induced. Smart hybrid composites are based on functional materials incorporated with fiber-reinforced polymers to benefit from multiple types of materials within one technical system. Shape memory alloys (SMAs) have received considerable attention as smart materials due to their exceptional strain recovery, good corrosion resistance, and strength in the fields of biomedical applications [1], and they are gaining much attention as smart functional materials in the aerospace industry for their structure, the control of aerodynamics in morphing structures [2–4], the noise reduction of aircraft engines [5], space research [6], robotics [7], sports [8], construction [9], and many engineering applications. Shape memory alloys undergo a reversible phase transformation that results in memorizing their original shape [10,11]. It is a phenomenon where they recover their original shape after deformation, by heating the materials above their transformation temperatures, known as the shape memory effect (SME), and the alloys are termed as

shape memory alloys (SMAs). The entire process is governed by four transformation temperatures, i.e., the martensitic start temperature $M_s$, martensitic finish temperature $M_f$, austenitic start temperature $A_s$, and austenitic finish temperature $A_f$, [12,13]. The transformation temperatures of these alloys can be controlled by controlling the composition and thermomechanical treatment, leading to a diversity of application areas at different temperatures.

A lot of research effort has been driven toward high-temperature SMAs [14,15]. In addition to smart behavior, these alloys also exhibit good mechanical properties that further increase their scope of utilization. SMAs exhibit another unique property termed superelasticity. Superelasticity (SE) is the property of materials that exhibits large elastic strains (4–8%) and recovers them upon removal of load at the same test temperature without a heating process [16]. As there is no requirement of temperature change and the process takes place in the isothermal regime, the material is termed as superelastic [13]. This effect is attributed to stress-induced martensitic transformation in these alloys upon application of stress/load, where they return to their original phase (austenite) upon removal of stress [17,18]. This property of energy absorption and dissipation can be utilized as impact absorbent material in a number of engineering applications. Equiatomic nickel titanium alloy (NiTi), also known as Nitinol (on account of its discovery at Naval Ordinance Lab USA), is an important SMA on account of its good strain recovery ability, good strength, and corrosion resistance [19]. It is one of the most widely used SMAs in different research areas.

In recent years, the fiber-reinforced polymers (FRPs) composite industry has rapidly been growing due to their light weight potential, good strength, and corrosion resistance, and they have been investigated by many research groups [20–23]. The weight efficiency along with good mechanical properties of FRPs make them efficient candidates for many engineering applications, especially in the transport industry for both automobiles and aircrafts where fuel efficiency is required. Research is also being conducted for the utilization of FRPs for the manufacturing of water transport vehicles [24]. The dependence of fuel consumption on weight is governed by the following equation [25]:

$$F = c_T \frac{C_D}{C_L} W \tag{1}$$

where $c_T$ is the specific fuel consumption, $W$ is the aircraft weight, $C_D$ is the drag coefficient, and $C_L$ is the lift coefficient. Thus, weight reduction has a direct influence on improving the fuel efficiency of the vehicle. Although GFRPs have a good strength-to-weight ratio, their impact properties are not very appreciable, as they undergo brittle failure under impact loading. The fibers cannot undergo plastic deformation unlike metals and they show brittle behavior. Reinforcement of polymer fiber composites with SMAs can, however, help to improve the energy absorption during an impact event as the superelastic NiTi can undergo phase transformation upon stress loading, leading to stress-induced martensite (SIM) transformation. Upon removal of load, the SMA returns back to the austenite phase. This whole process can be understood by the energy hysteresis during the loading and unloading cycle, which shows the energy absorbed during the process, as shown schematically in Figure 1.

Reinforcing GFRPs with SMAs is expected to improve the impact properties of the hybrid composite material; thus, shape memory wires can overcome the brittle nature of GFRPs. The SMAs can absorb the energy of impact due to their hysteretic nature as depicted in Figure 1. SMAs can improve impact damping in both shape memory and superelastic forms; however, in the case of superelasticity, reverse transformation will take place without heating. For the case of a shape memory alloy in the martensitic phase, there are glissile interfaces that can move easily, so that impact energy can be absorbed, leading to energy damping [26] and improving the impact resistance of the composite material. However, the impact resistance of SMAs in the martensitic phase is found to be temperature-dependent [27]. Superelastic SMAs can improve the damage tolerance and impact strength of the GFRPs in the low- and medium-velocity regime due to excellent

super-elastic deformation and shape recovery properties [28,29]. Guida et al. [30] worked on the integration of SMA wires in carbon fiber thermoplastic composites, and showed a higher toughness and absorption of impact energy due to its superelastic and hysteretic properties.

This idea is further supported by material property maps of Ashby, which show that SMAs possess high actuation stress, strain, and axial stiffness [31,32]. SMA-based actuators have found many applications in the research industry [33,34]. On the other hand, GFRPs possess good actuation behavior at low mass density. Thus, the reinforcement of GFRPs with SMAs can benefit from good mechanical and actuation properties of both materials at low weight, thus improving the weight efficiency of the resultant composite material, best suited for the transport and aerospace industry.

The goal of this research work is focused on the development of SS-reinforced GFRPs and SMA-reinforced GFRP hybrid composites for improving the impact properties of the FRP composite materials that suffer from impact damages during processing and in-service impact events, and to make a comparison of the effect of metallic alloy reinforcement (stainless steel) vs. superelastic (NiTi) reinforcement that undergoes stress-induced pseudoelastic deformation before plastic deformation and, thus, absorbs the energy of impact. Stainless steel (SS) being a metallic alloy undergoes plastic deformation upon loading and can, thus, improve the load-bearing capacity of the composites, and it is used for reinforcement, especially in the construction field in combination with cement [35], whereas superelastic wires can absorb sufficient energy of impact that can be later dissipated, due to energy hysteresis [29,30]. This helps in decreased damage and greater energy absorption before failure. In addition, the SMA-reinforced hybrid composite exhibits a recentering ability that helps to reduce damage upon exposure to stress [36,37]. In this research work, superelastic SMAs in the form of a mesh of wires are integrated in conventional glass fiber composites and the energy absorbed on impact/toughness, impact strength, and specific strength are analyzed and compared using the Charpy impact test. Such research can be utilized in the transport industry such as automobiles and aircrafts that can help to improve the impact resistance of the vehicle during processing in industry and in in-service events. In addition, such hybrid composites have a better specific strength that makes them ideal as weight-efficient materials required for the transport industry for fuel saving.

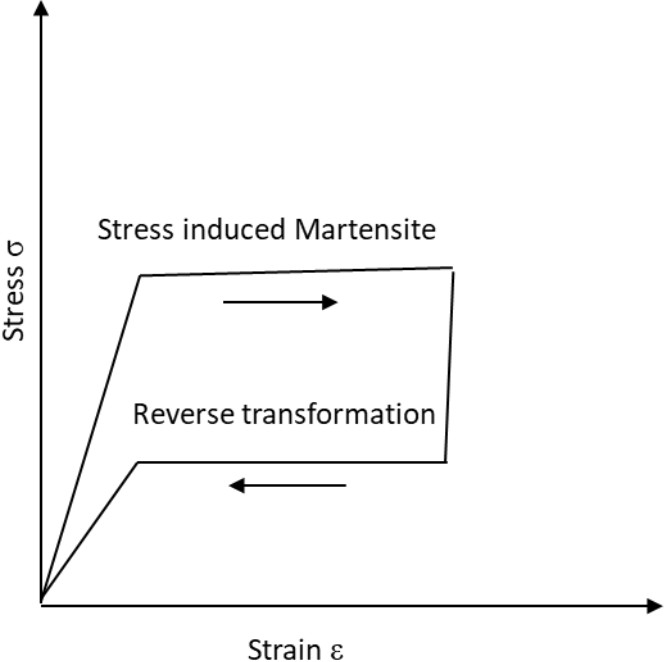

**Figure 1.** Reversible transformation from austenite to stress-induced martensite phase of superelastic SMA at test temperature $>A_f$, showing energy hysteresis.

## 2. Materials and Methods

Superelastic NiTi near-equiatomic alloys in the form of wires of diameter 0.3 mm and stainless-steel wires of diameter 0.3 mm, obtained from Ortho Organizers USA, were used in the present research work to manufacture SMA-reinforced FRP hybrid composites and SS-reinforced FRP hybrid composites. Tensile testing of NiTi wires was conducted to analyze their stress vs. strain response. E glass fibers were used as FRPs to manufacture glass-fiber-reinforced composites and in hybrid combination with SMAs and SS. The composition of fibers as obtained from EDX analysis is shown in Table 1. The experimental steps followed are shown in Figure 2 and represented schematically in Figure 3.

1. Preparation of matrix (epoxy resin + hardener, 1:4)

2. Degassing of resin mixture

3. Treatment of wires followed by preparation of wires mesh

4. Hand layup of glass fiber + SMA/stainless steel wires + Epoxy

5. Vacuum bagging of laminate

6. Compression molding of composite for 24 hrs + 2 hrs furnace curing at 80 °C

**Figure 2.** Manufacturing sequence of hybrid SMA/FRP composite.

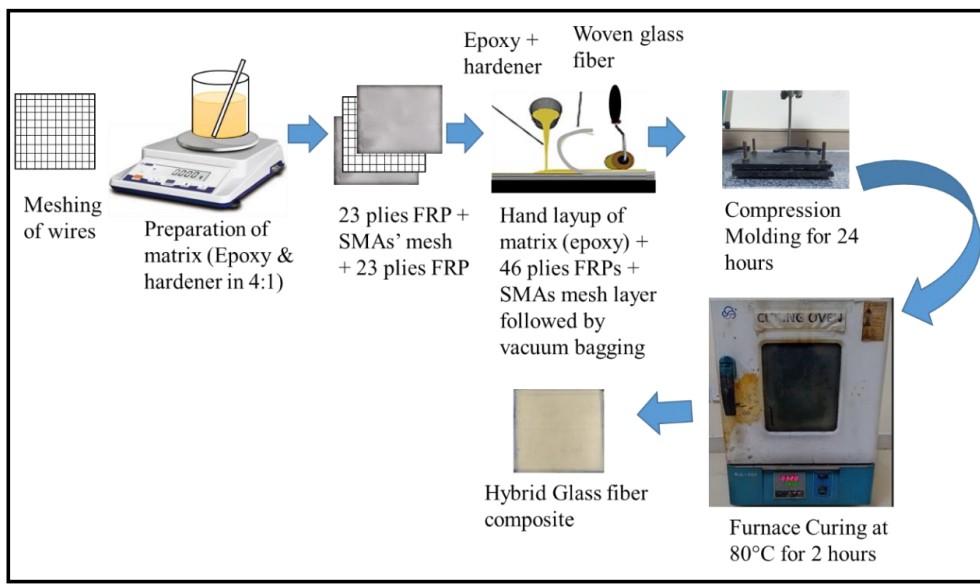

**Figure 3.** Schematic of experimental sequence for composite manufacturing.

Both stainless-steel and NiTi superelastic wires were treated mechanically to improve adhesion with the matrix. The treatment included abrasion with emery paper along the horizontal direction to the length of wires with 220-grit paper, followed by abrasion with 400-grit emery paper. After mechanical grinding, wires were kept in ethanol with ultrasonification for 20 min followed by ultrasonification in deionized water for 20 min. The mechanical treatment was carried out as reported in the patent [38] by Okonski et al. Treated wires were fabricated as a mesh to be integrated in the composite with a volume fraction of wires of 8%, as shown in Figure 4.

**Table 1.** EDX composition of E glass fiber.

| Element | Weight % | Atomic % |
|---------|----------|----------|
| B K | 13.13 | 20.10 |
| C K | 13.02 | 17.94 |
| O K | 45.07 | 46.61 |
| Na K | 0.45 | 0.32 |
| Al K | 4.12 | 2.53 |
| Si K | 14.21 | 8.37 |
| Ca K | 9.99 | 4.13 |
| **Total** | 100.00 | 100.00 |

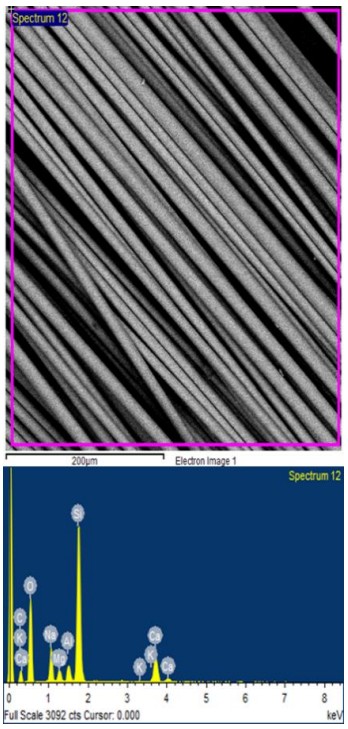

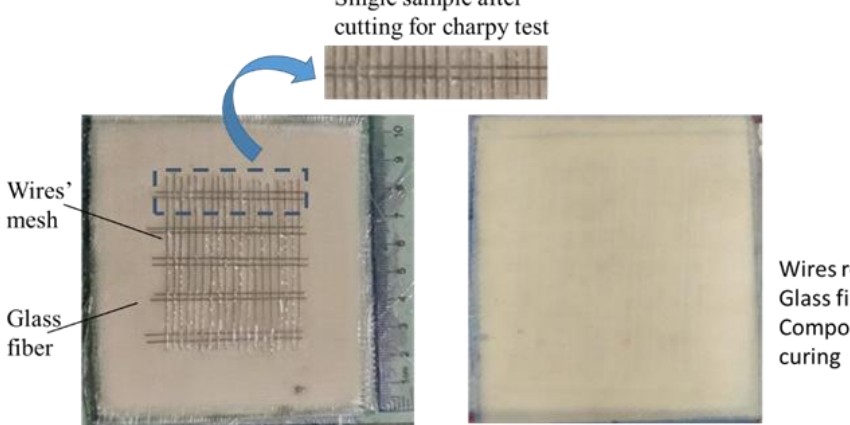

**Figure 4.** Glass fiber with wires' mesh.

The wires' mesh was integrated in the mid plane of woven glass fiber fabric laminate consisting of 46 plies. We used commercial-grade epoxy as a matrix (with hardener-to-epoxy ratio of 1:4). After stirring the epoxy, degassing of the resin and hardener mixture was carried out using a vacuum pump to remove bubbles. The laminate was fabricated using a hand-layup technique followed by vacuum bagging (using rotary pump) and compression molding. The compression-molded laminate was cured at room temperature for 24 h, followed by furnace-curing at 80 °C for 2 h. After curing, the sample was cut for the Charpy impact test with dimensions of 6 mm × 6 mm × 44 mm in accordance with the MT 3016 test specimen requirement. Three types of composites were manufactured: the glass-fiber-polymer-reinforced composites (GFRPs), stainless-steel-reinforced glass fiber composites (SS/GFRPs), and the NiTi-superelastic-alloy-reinforced glass fiber composites (SMA/GFRPs), also named as smart hybrid composite. A second set of experiments were carried out and test samples were manufactured with dimensions of 6 mm × 6 mm × 57 mm (set B) for better gripping in the holder of the Charpy impact tester. Figure 5 shows the impact tester used for the Charpy impact test. It is a 3016 MT impact tester, which is a robust bench impact tester.

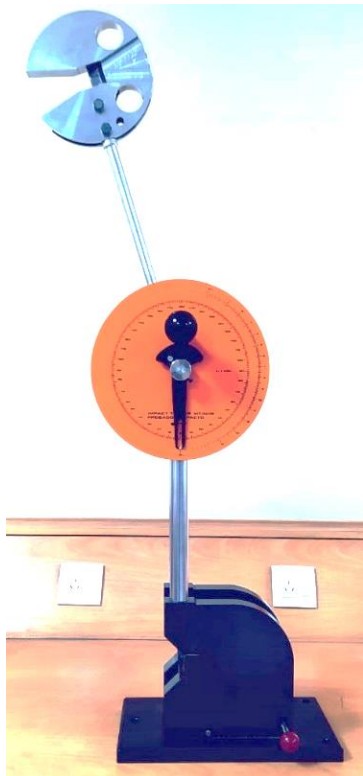

**Figure 5.** Charpy impact testing machine used for impact testing of composite samples.

### 3. Results and Discussion

Optical micrographs of the untreated SS wire surface, mechanically treated and ultrasonicated SS wire surface, untreated SMA wire surface, and mechanically treated and ultrasonicated wire surface are shown in Figure 6. The optical microscope used was a Novex Holland with a magnification 1000×. It can be observed that mechanical treatment substantially improved the surface roughness of the wires.

Figure 7 shows optical microscopy of the cross-section of the SS and SMA FRP hybrid composites, which showed good adhesion between the wires and FRPs at the interface.

The Charpy impact test results for the first set of experiments with dimensions of 6 mm × 6 mm × 44 mm (set A) are shown in Table 2.

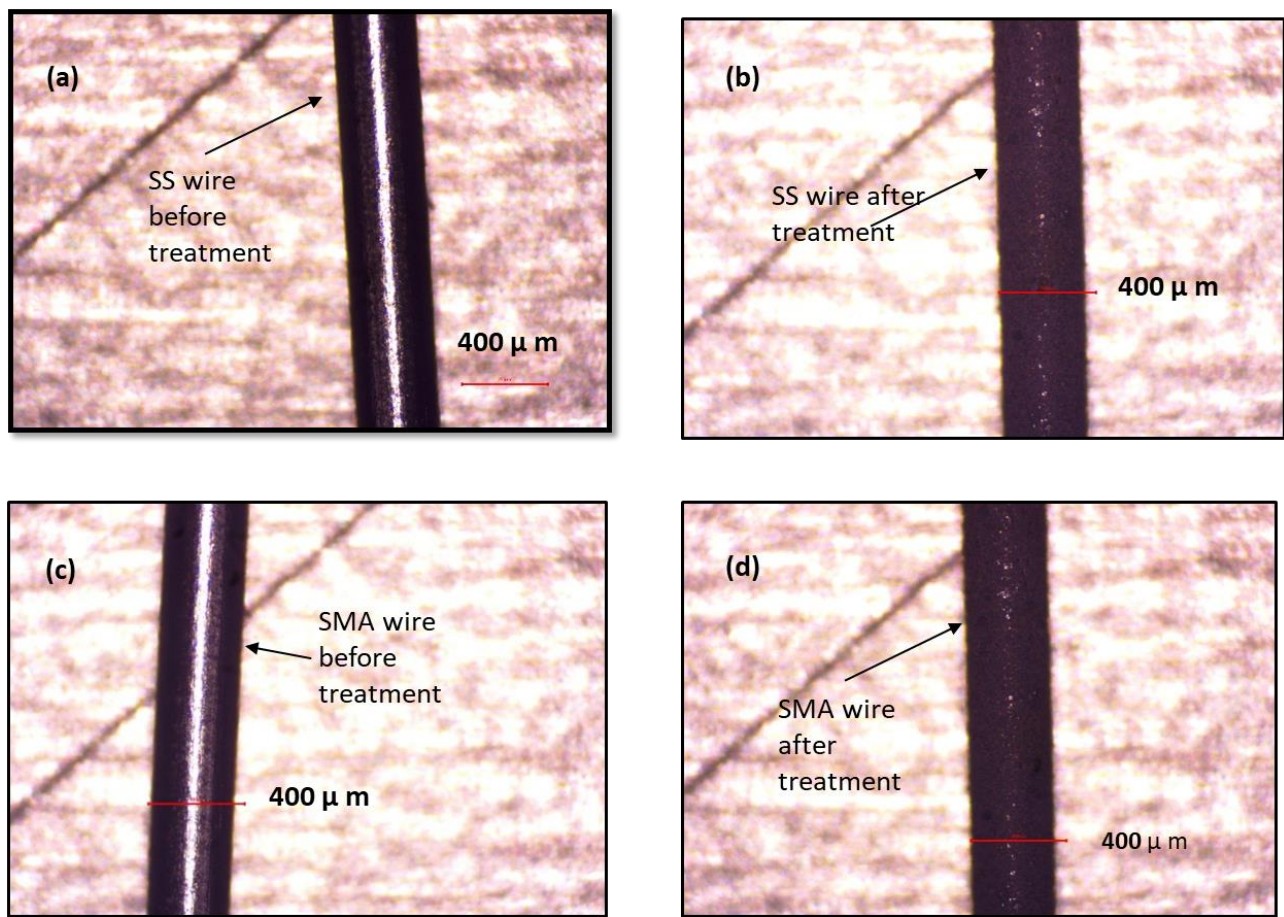

**Figure 6.** Optical micrographs of (**a**) untreated SS wire surface, (**b**) mechanically treated and ultrasonicated SS wire surface, (**c**) untreated SMA wire surface, and (**d**) mechanically treated and ultrasonicated wire surface.

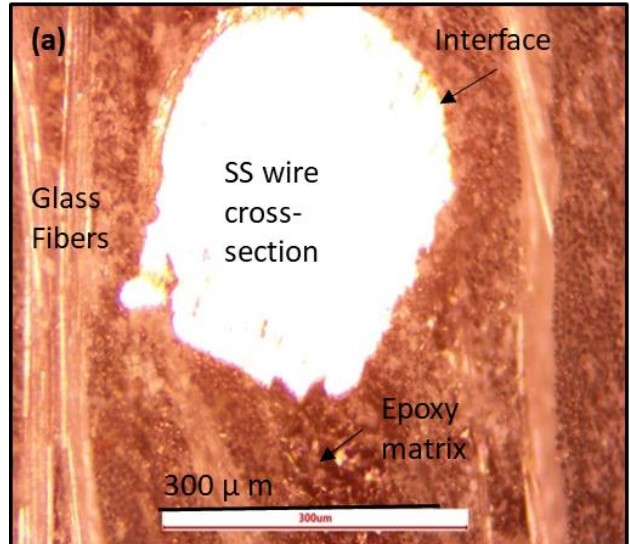 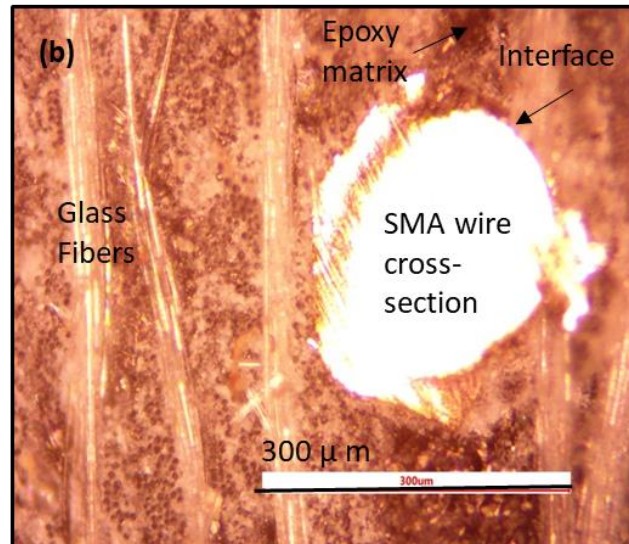

**Figure 7.** Optical micrographs of cross-section of hybrid composites after treatment of wires: (**a**) SS/GFRP hybrid composite; (**b**) SMA/GFRP composite.

**Table 2.** Results of Charpy Impact Test set A.

| Scheme. | Composite Sample | Energy Consumed in Breaking (J) | Impact Strength (kJ/m$^2$) |
|---|---|---|---|
| 1. | GFRP | 8 | 258 |
| 2. | SMA/GFRP | 7 | 217 |
| 3. | SS/GFRP | 5 | 149 |

GFRP—Glass fiber reinforced polymer. SS/GFRP—Stainless steel mesh reinforced GFRP. SMA/GFRP—Shape memory mesh reinforced GFRP.

The energy absorption capability and impact strength of hybrid stainless-steel (SS)-reinforced GFRP and hybrid-SMA-reinforced GFRP was less than those of the GFRP composite, as shown in Figures 8a and 9a, respectively. It was analyzed that the samples with wires were bent and slipped out of the grips of tester. Samples should not slip out of the grips of the tester. As samples slipped out from the grips of the tester, the results reported in Table 2 are not accurate. Although GFRPs samples absorbed sufficient impact energy, during the test, however, stainless-steel (SS) and shape memory wires samples did not absorb sufficient energy and were slipped along with the tester hammer, during this test. Thus, these samples with SS and shape memory wires absorbed less energy than GFRPs. We analyzed that the length of sample was small and there was only a small grip area of samples, that is, under the grips of sample holder.

The second set of experiments with dimensions of 6 mm × 6 mm × 57 mm (set B) was carried out by increasing the length of the composite by 6.5 mm on both sides of the grip to avoid slipping from the grips. The Charpy impact test results for the second set of experiments with dimensions of 6 mm × 6 mm × 44 mm (set B) are shown in Table 3.

The absorbed energy profile/toughness of the material for the glass-fiber-reinforced composites (GFRP), SMA-reinforced GFRP (SMA/GFRP), and stainless-steel-wires-reinforced GFRP (SS/GFRP) is shown in Figure 8, and the impact strength of all the samples is shown in Figure 9. Impact test results of set B are shown in Table 3 and in Figures 8b and 9b. For set B, the SMA/GFRP hybrid composite showed a higher impact strength than SS/GFRP. This is attributed to the stress-induced martensitic transformation of SMAs upon impact loading, shown by the plateau region of the tensile stress–strain curve of the SMA wire (Figure 10a). Figure 10b shows the Shimadzu universal testing machine used for tensile testing of NiTi SMA wires. Increasing the length of specimens was found to be effective in improving the gripping of the composite, which led to improved energy absorption capability and impact strength of the hybrid composites as compared to GFRPs, as shown in Figures 8b and 9b, respectively. The results were analyzed with respect to three boundary conditions followed by fracture analysis.

**Table 3.** Results of Charpy Impact Test set B.

| Sr. No. | Composite Sample | Energy Consumed in Breaking (J) with Notch | Energy Consumed in Breaking (J) without Notch | Impact Strength with Notch (kJ/m$^2$) | Impact Strength without Notch (kJ/m$^2$) |
|---|---|---|---|---|---|
| 1. | GFRP | 7.8 | 13 | 229 | 361 |
| 2. | SMA/GFRP | 11 | 14.6 | 323 | 406 |
| 3. | SS/GFRP | 10 | 13.5 | 294 | 375 |

GFRP—Glass fiber reinforced polymer. SS/GFRP—Stainless steel mesh reinforced GFRP. SMA/GFRP—Shape memory mesh reinforced GFRP.

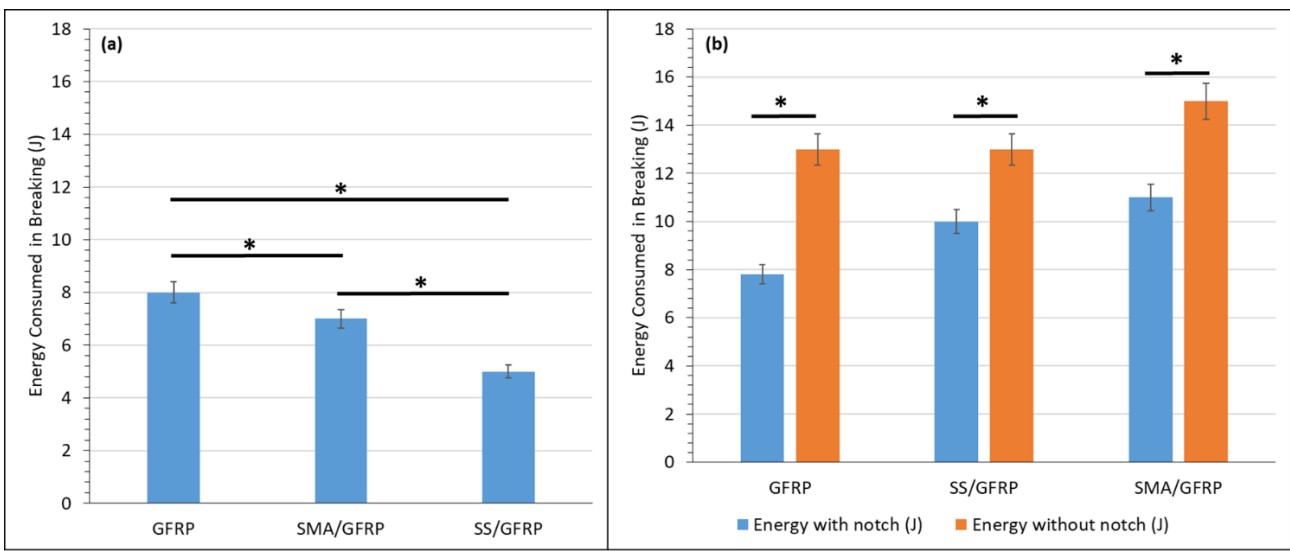

- GFRP − Glass fiber reinforced polymer
- SS/GFRP − Stainless steel mesh reinforced GFRP
- SMA/GFRP − Shape memory mesh reinforced GFRP

**Figure 8.** Results of Charpy impact test indicating energy consumed in breaking (toughness) of GFRP, SS/GFRP, and SMA/GFRP composites: (**a**) set A, with dimensions of 6 mm × 6 mm × 44 mm; (**b**) set B, with dimensions of 6 mm × 6 mm × 57 mm. At a *p* value of 0.05, the energy consumed in breaking is significantly different for all samples. The difference in the consumed energy with and without the notch is also significant. * symbolizes the significant difference.

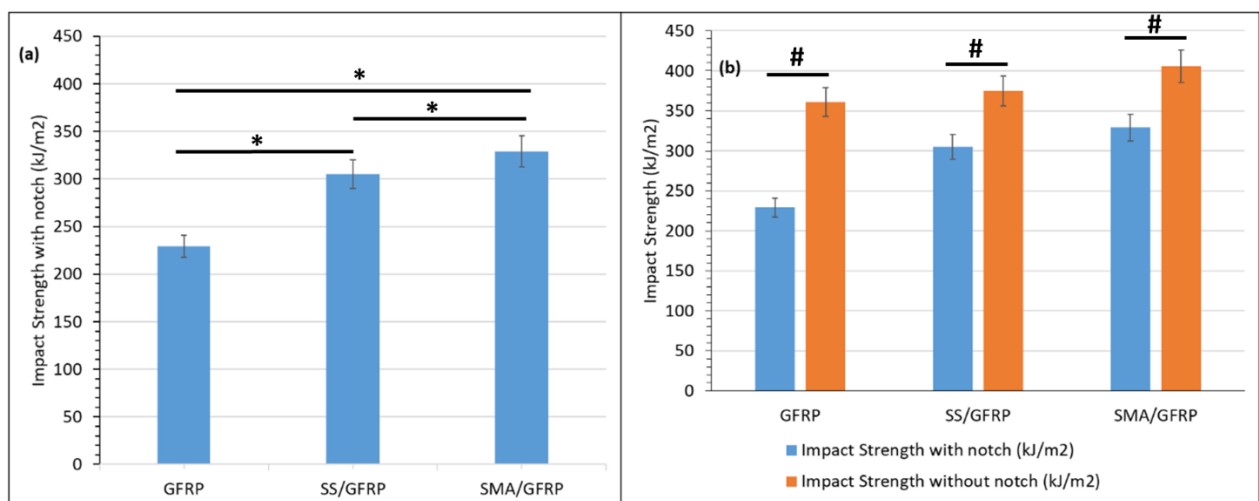

- GFRP − Glass fiber reinforced polymer
- SS/GFRP − Stainless steel mesh reinforced GFRP
- SMA/GFRP − Shape memory mesh reinforced GFRP

**Figure 9.** Results of Charpy impact test indicating impact strength for GFRP, SS/GFRP, and SMA/GFRP composites: (**a**) set A, with dimensions of 6 mm × 6 mm × 44 mm; (**b**) set B, with dimensions of 6 mm × 6 mm × 57 mm. At a *p* value of 0.05, the impact strength with the notch is significantly different for all samples. However, the difference in impact strength with and without the notch is not significant. * symbolizes the significant difference and # symbolizes the insignificant difference.

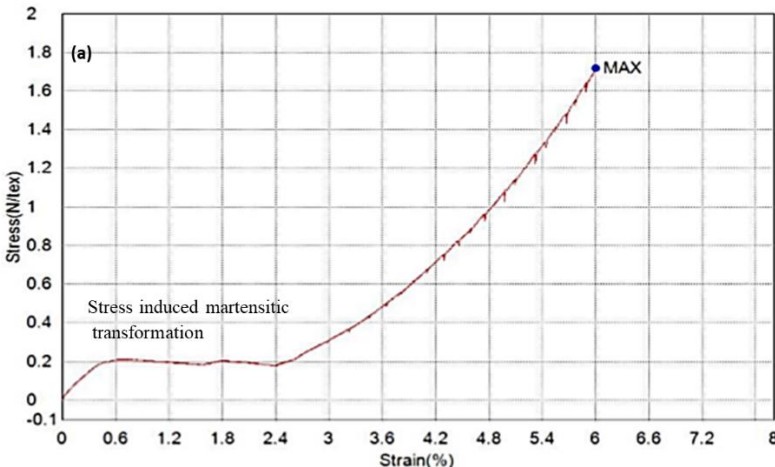
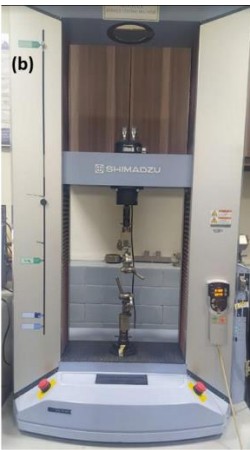

**Figure 10.** (**a**) Tensile stress-strain curve of NiTi wire produced until 6% strain, showing pseudo-elastic deformation due to stress-induced martensitic transformation; (**b**) Schimadzu Universal testing machine used for tensile test.

### 3.1. Boundary Conditions

#### 3.1.1. Effect of Dimensions

The first set of samples was prepared with dimensions of 6 mm × 6 mm × 44 mm (set A) according to the MT 3016 standard, while the second set of composites were manufactured with dimensions of 6 mm × 6 mm × 57 mm (set B). The results of impact tests with both dimensions are shown in Tables 2 and 3, respectively. It can be observed that increasing the length of composites resulted in better impact energy absorption i.e., better toughness, due to better gripping of the sample in the tester holder (Figure 8). The smaller samples were not properly gripped and slipped along with the tester hammer. No appreciable damage or deformation of the wires for the smaller length samples (set A) could be observed, as evident from the stereo fractography. The micrography of the samples after impact damage was conducted using a stereomicroscope, Nikon SMZ25 (Model P2-Firl). It has a unique zoom range of 25:1 and is widely used for fracture analysis worldwide. Figures 11–13 show the cross-sectional and top view stereomicrographs of set A composites. The glass fibers were damaged by impact but insufficient gripping led to the elastic bending of wires and slip of composites with a tester hammer.

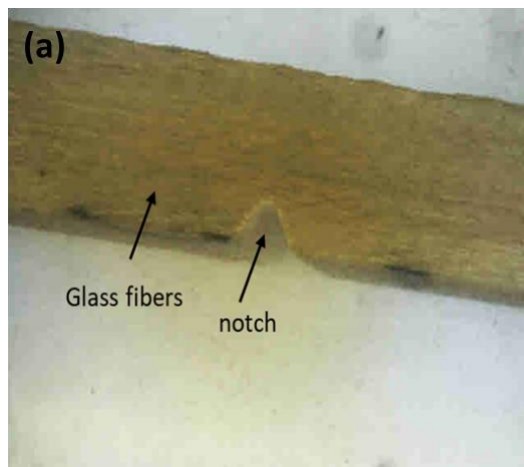
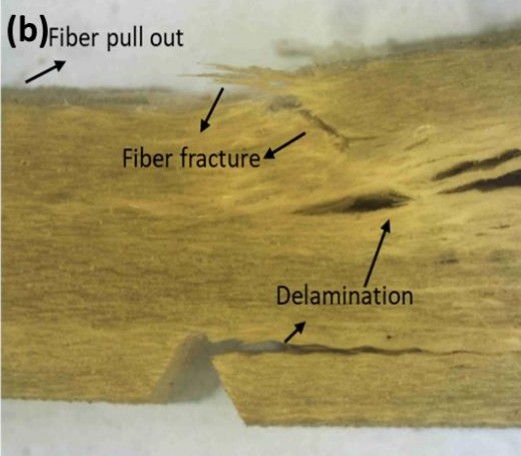

**Figure 11.** Set A: (**a**) GFRP composite before impact; (**b**) GFRP composite after impact.

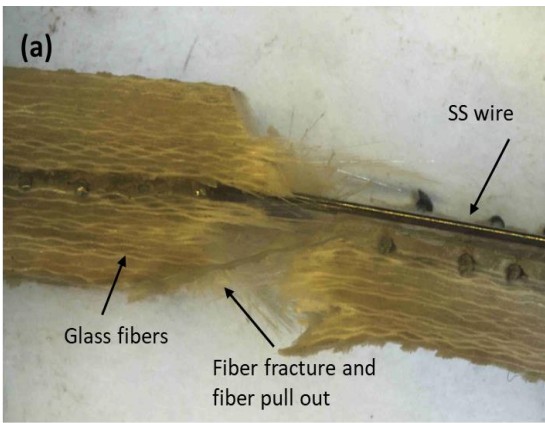
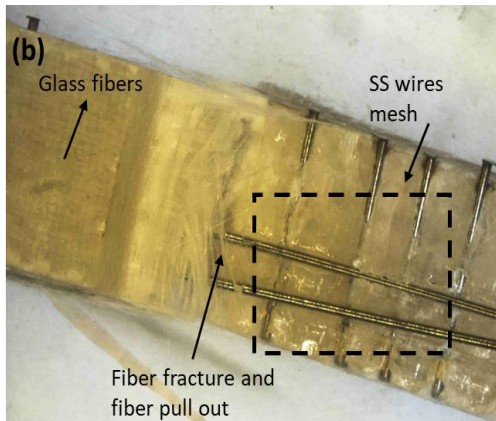

**Figure 12.** Set A: (**a**) cross-sectional view of SS/GFRP composite after impact; (**b**) top view of GFRP composite after impact.

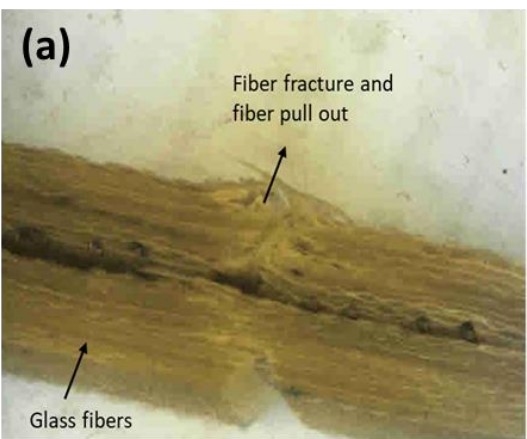
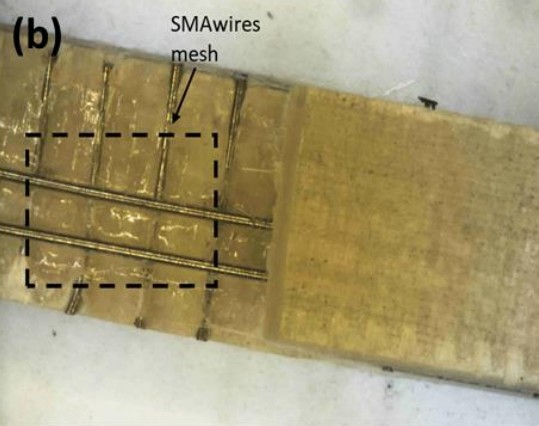

**Figure 13.** Set A: (**a**) cross-sectional micrograph of SMA/GFRP composite after impact; (**b**) top view of SMA/GFRP composite after impact.

However, an increasing length of samples resulted in a better gripping of the samples in the tester holder (Figures 14–16, set B).

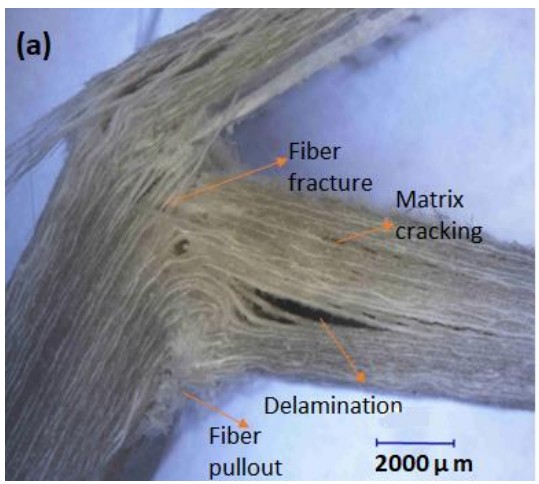
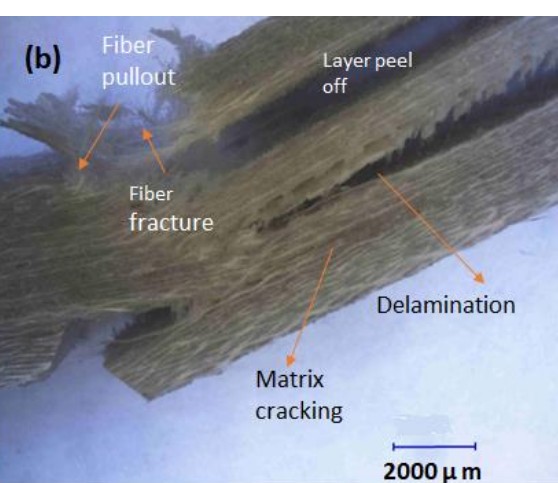

**Figure 14.** Set B: (**a**) unnotched GFRP composite; (**b**) notched GFRP composite.

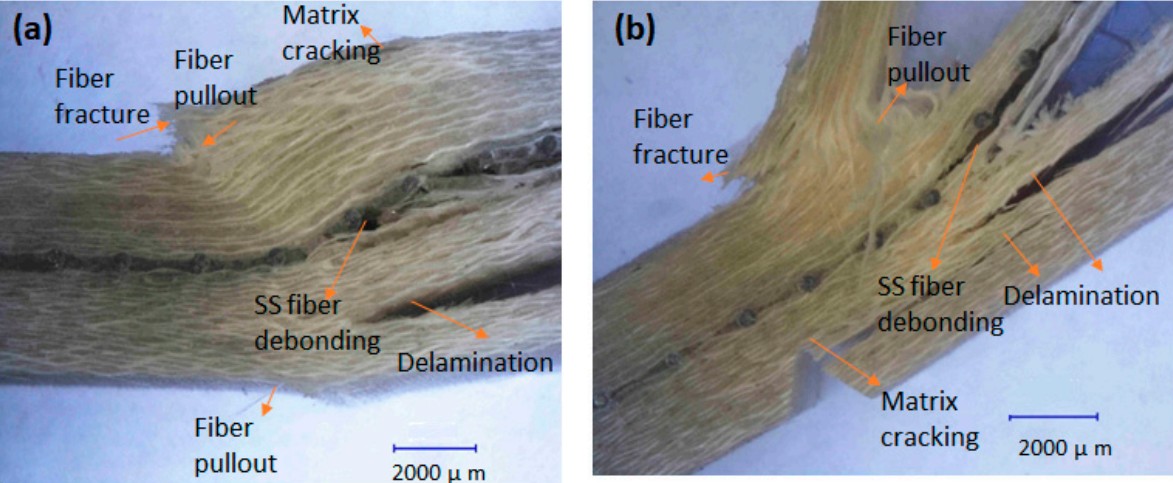

**Figure 15.** (**a**) Unnotched SS/GFRP; (**b**) notched SS/GFRP.

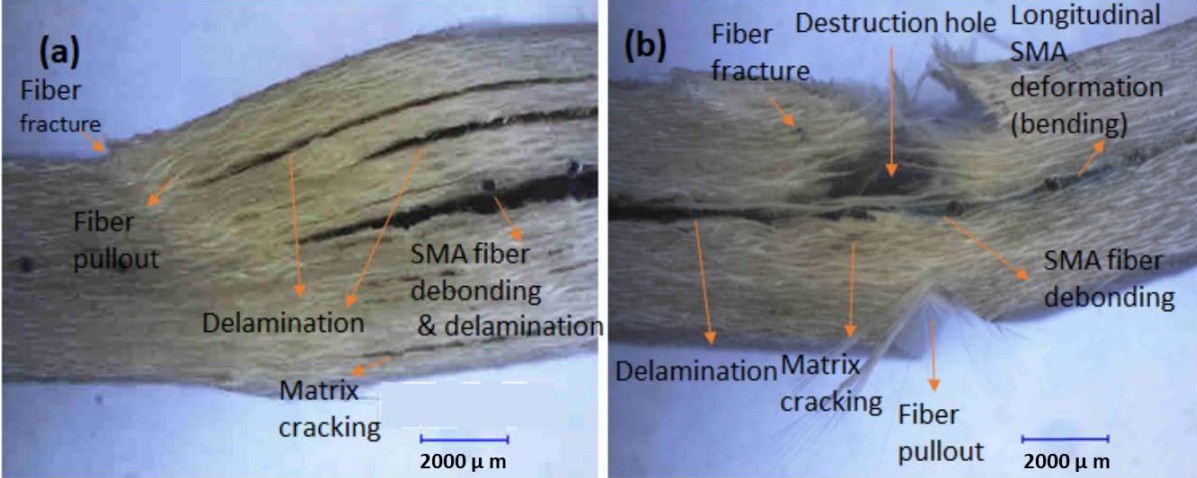

**Figure 16.** (**a**) Unnotched SMA/GFRP; (**b**) notched SMA/GFRP.

The samples of set B were affected by the impact, resulting in deformation and fracture. In set B also, wires were deformed, showing impact energy absorption unlike set A that hardly showed any effect of impact on wires. Various fracture mechanisms were observed in set B, i.e., matrix cracking, delamination, fiber pullout and fiber fracture, and wires deformation, as depicted in the optical micrographs. In Section 3.1.2, the effect of the notch was also determined for set B, which is discussed below.

### 3.1.2. Effect of Notch (Set B)

To study the effect of the notch on the fracture mechanics and energy absorption during impact, specimens were prepared with and without the notch. The notch acts as a stress generator region and provides a crack initiation point, and the strength of the composites with flaws can be computed. It helps to study the influence of flaws on the impact strength of composite materials. These defects can reduce the strength of the composites and affect their performance. Thus, the impact strength is determined with notches to study the fracture toughness to predict the life of materials in service with in-built flaws. In addition, the ASTM standards compute impact damages with notches. The impact strength of the composite specimens without the notch shows the total energy absorbed without any stress generator, so the crack propagation path is not defined. However, the flaws in the composite or interfaces may act as stress generators in this case. In a study, Backlund [39] introduced a Damage Zone Model for notched composite laminates using finite element

analysis to analyze the stress peak at the notch and damage at the stress region, i.e., the edge of the notch. Eriksson et al. also developed a Damage Zone Criterion model based on the assumption that a homogeneous damage zone grows at the notch tip, perpendicular to the direction of force [40,41]. This fracture profile can be observed in the cross-sectional fracture images in Figure 11b, Figure 12a, Figure 14b, and Figure 15b, respectively. It can also be observed from Figures 14–16 that the fracture mode for unnotched composites was mostly delamination, while for the notched composites, fiber fracture and fiber pullout were dominant along with the delamination propagation. Thus, more fracture mechanisms were activated in the presence of the notch due to stress concentration at the notch tip.

From the experimental results, it can be observed in Figures 8b and 9b that the samples without the notch absorbed more energy of impact before failure as compared to the samples with the notch.

This study is important to determine the impact strength of the materials with and without the notch to estimate the lifetime of in-service composites, those without flaws or stress generators, and those with notches. Thus, the in-service impact strength of composites having barely visible impact damage (BVID) can be conceptualized. Baluch et al. investigated the damage tolerance of laminates for BVID [42]. Defects, minor cracks, and damages can occur in composites during manufacturing or in-service performance [43]. We observed from our results that the notched specimens had less impact strength as compared to the unnotched specimens. Thus, the loaded composites, i.e., those having a notch or BVID during processing, resulted in a decrease in inherent impact strength and toughness of the material and underwent different fracture mechanisms and a higher extent of damage on impact events [44].

### 3.1.3. Effect of Wires (Set B)

In order to improve interfacial bonding between the wires and the matrix, the wires were treated to improve their roughness, to increase their adhesion with the matrix to lead to better interfacial strength. Mechanical treatment helps to increase the roughness and, hence, surface area of the wires, enabling better mechanical interlocking between the wires and the matrix [45]. Better interfacial adhesion can impart strength to the resulting composite material. Figure 6 shows an optical micrograph of the surface of SS and SMA wires before and after treatment. It can be observed that the roughness of the wire appreciably improved. The treated wires, both SS and SMA, were used as reinforcement in the hybrid composites. The samples after fabrication and cutting were analyzed for bonding under an optical microscope. The micrograph of the cross-section of samples is shown in Figure 7. As the micrographs show, the wires were well-bonded to the matrix and laminate. The samples were tested for impact energy absorption, i.e., toughness and impact strength, using the Charpy impact test. The results obtained are shown in Table 2 and graphically represented in Figures 8b and 9b. As can be observed from the fractographs, the wires in the transverse direction to the length of the composite were well bonded to the matrix, while those in the longitudinal direction were affected by the impact. Thus, it was supposed that transverse wires give structural integrity to the composite and the longitudinal wires directly absorb impact energy. Both SS and SMA wires reinforcement were found to improve the toughness and impact strength of the hybrid composites as they were ductile and underwent plastic deformation, and, thus, could overall improve the brittle behavior of FRPs. In addition, the various fracture mechanisms at the interface such as delamination and fiber pullout improved the overall energy absorption during an impact event.

It can be observed from Figure 9b, for the unnotched composites, that the stainless-steel wires' mesh had an increased impact strength of the hybrid SS/GFRP composite as compared to the glass fiber composite by 4%. Thus, steel wires were good reinforcement to improve the toughness of the material due to their greater strength. Metallic alloys also underwent plastic deformation upon loading, thus absorbing energy rather undergoing fracture, allowing improved toughness of the composites. In addition, the SMA/GFRP sample increased the impact strength of the glass fiber composite by 13%. This remarkable

increase in impact strength of the smart SMA glass fiber composites is attributed to the energy absorption capability of the superelastic wires owing to their energy hysteresis that absorbs the impact load, leading to stress-induced martensitic transformation followed by plastic deformation [46,47].

For the samples with the notch, the SS increased the impact strength of GFRP by 28%, while the SMA/GFRP increased the impact strength of glass fiber by 41%.

### 3.2. Fractography

After impact, the stereomicrographs of the fractured samples are shown in Figures 11–16. Figures 11–13 show micrographs of set A samples. Figure 11a shows the GFRP sample before impact and Figure 11b shows the GFRP sample after impact testing. The impact damage included fiber fracture, fiber pullout, and delamination. An oblique crack can also be observed from Figure 11b. Figure 12a,b show cross-sectional and top views of the impact-damaged sample of the SS/GFRP composite. The glass fiber was fractured and a layer was peeled off. The wires were not fractured. Only minor deformation can be observed in wires along the longitudinal direction of the composite. As the size of the sample was small, there was insufficient gripping to pass the stress to the wires and allow sufficient energy absorption. The wires elastically bent (followed by a little plastic deformation), and the sample moved out with the tester hammer. The same effect can be observed in Figure 13a,b with SMA/GFRP composites. Figure 13a shows the cross-sectional view of the SMA/GFRP composite and Figure 13b shows the top view of the composite after impact testing. It can be observed that glass fibers were damaged via fiber fracture and pullout, the fiber layer peeled off, but the SMA wires were not damaged. Being superelastic, they underwent pseudo-elastic-stress-induced martensitic transformation and returned to their original configuration after impact, as shown in Figure 1, and because the sample size was small, there was not enough time for them to deform plastically before the sample moved out with the tester hammer. Due to superelastic nature of wires, the sample was in the straight configuration and was not even bent after impact. Thus, SMAs help in damage reduction after impact. However, to observe the actual toughness and impact strength of the smart hybrid composites, the length of the samples should be sufficient to allow sufficient gripping and, thus, to observe the toughness that can be induced in composites.

In the composites, as can be observed from micrographs in Figure 14b, Figure 15b, and Figure 16b, the notch provided a stress concentration area; the fracture propagated from the cross-sectional area around the notch, perpendicular to the hole; delamination of fibers can also be observed for all samples within the cross-section of the notch as compared to the unnotched samples. Thus, the absorbed energy and impact strength were lower for the notched samples as compared to the unnotched samples. The energy absorbed was maximum for SMA-reinforced GFRP samples. Here, the impact fractured the fibers until the fracture propagated to the plane of super-elastic wires that absorbed the impact energy, leading to stress-induced martensitic (SIM) transformation and absorption of energy required for transformation, followed by plastic deformation. The SMAs underwent double yielding upon application of stress and, hence, could allow sufficient damping of energy. This can be observed from Figure 16. The wires absorbed maximum energy and the fracture was not dominant in the composite on the bottom side of the plane of superelastic layer. After SIM, plastic deformation of the super-elastic wires took place, leading to the bending of wires macroscopically, as can be observed in the stereofractographs in Figures 14–16. The SIM can be observed in the plateau region of the tensile stress–strain plot of NiTi wire, as indicated in Figure 10a. Thus, superelastic SMAs improved overall the structural integrity of the reinforced hybrid composites owing to their smart behavior. In the application industry, they can bear and recover small stresses owing to pseudoelastic SIM transformation and their recentering ability and, thus, can help to improve the overall damage resistance.

The stainless-steel wires also provided strength to the composite due to the load transfer capability and they had greater strength as compared to glass fibers, and the

damage in SS-reinforced GFRP composites in Figure 15 was not as drastic as the GFRP composite (Figure 14). Thus, the resulting hybrid composite SS/GFRP had greater impact resistance as compared to the GFRP composite.

The unnotched composites, i.e., GFRP composites, the SS-reinforced hybrid GFRP composite, and the smart hybrid composite SMA/GFRP, all absorbed more impact energy as compared to the notched samples as there was no stress concentration area around the fracture. For the glass fiber composite, Figure 14a shows that the load was transferred throughout the cross-section of the composite, leading to a V-shaped morphology of the cross-section. The damages comprised matrix breakdown, fiber bending, fiber breakage, and multilayer delamination, all of which showed energy dissipation mechanisms within the composite, but the sample was not broken as was the case of the notched sample. For the SS-reinforced GFRP unnotched sample, Figure 15a, the stainless-steel mesh provided strength to the hybrid composite, propagated as fiber bending and breakage at the point of impact, and delaminated in the fibers and in the plane of the SS mesh. The absorbed energy was greater than that of the glass fiber composite, leading to a greater impact strength of the material. For the case of the SMA-reinforced glass fiber composite, unnotched sample (Figure 16a), the impact energy absorbed was greatest among all composite samples, leading to a greater impact strength of the smart hybrid composite. The damage after impact consisted of matrix breakdown and fiber delamination in different planes including the plane of wire. The wires were plastically deformed after SIM transformation upon absorbing impact energy, but there was no fracture of wires, as shown in Figures 15 and 16. The absorbed energy was due to energy hysteresis of the superelastic shape memory wires that provided overall a greater impact strength to the composite material. The plastic deformation observed in SS wire was greater in the SS-reinforced GFRP composites as compared to SMA wires as SMAs underwent phase transformation to stress-induced martensite, which can be observed in the plateau region of the stress–strain curve of SMAs, in Figure 10. Thus, SMA hybrid composites improve the damage resistance of the composites, as discussed in [48].

From these results, it is clear that the impact strength of the SMA-reinforced hybrid GFRP composite had the highest value, making it the most impact-resistant material.

Figure 17 shows a plot of the specific strength of the impact test of the GFRP, SS/GFRP, and SMA/GFRP composite, and the SMA/GFRP composite showed the highest value as predicted, making it the most weight-efficient composite against impact. At a *p* value of 0.05, specific impact strength was significantly different for all samples. * symbolizes the significant difference.

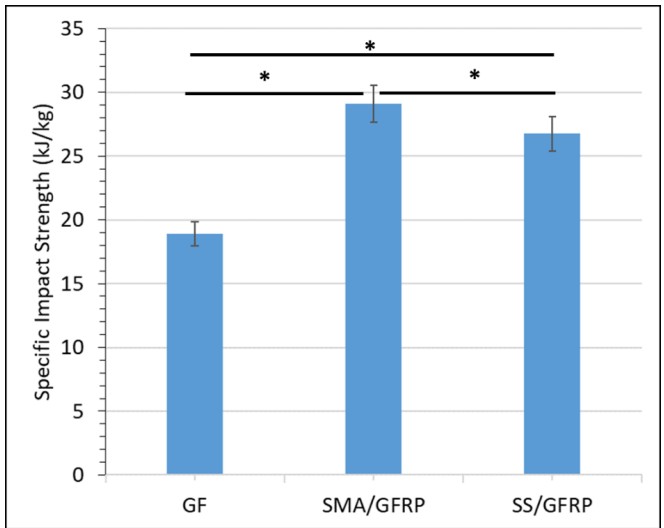

**Figure 17.** Specific strength of GFRP, SMA/GFRP, and SS/GFRP composites. At a *p* value of 0.05, specific impact strength is significantly different for all samples. * symbolizes the significant difference.

## 4. Conclusions

We successfully developed GFRP, SS-reinforced GFRP, and SMA-reinforced GFRP hybrid glass fiber composites.

Increasing the length of the impact sample from 44 mm to 57 mm resulted in better gripping of the sample in the holder of the impact tester, avoiding slippage and leading to improved energy absorption during impact.

The unnotched composites had greater toughness and impact strength as compared to the notched samples as the fracture strength without the flaw/notch was greater than the fracture strength with the flaw/notch, where the notch acted as a stress concentration point, responsible for the crack propagation path.

The impact strength of the SS/GFRP hybrid composite was 4% greater than that of the unreinforced GFRP, the impact strength of SMA/GFRP hybrid composite was 13% greater than that of the unreinforced glass fiber composites, while the toughness of the SS/GFRP hybrid composite was 3.8% greater than that of the unreinforced GFRP, and the toughness of the SMA/GFRP was 12.3% greater than that of the unreinforced GFRP.

The damage resistance of the SMA-reinforced hybrid composite was greater than that of the SS-reinforced hybrid composites as evident from the stereofractographs.

The specific impact strength of the SS/GFRP hybrid composite was greater than that of the unreinforced GFRP composite by 41% and the specific impact strength of the SMA/GFRP hybrid composite was greater than that of the unreinforced GFRP composite by 53%. Thus, the specific impact strength of the smart superelastic hybrid glass fiber composite (SMA/GFRP) was highest among all composites tested, making it the most weight-efficient composite material.

**Author Contributions:** The authors' contributions are as follows: H.O. conceptualized, planned, carried out the experiments, and prepared the original draft. A.H.B. contributed to the analysis and interpretation of the article. M.A.U.R. validated and edited the original draft. A.W. supervised and critically reviewed the research and manuscript. I.Q. contributed in the review and design of experiments. K.Y. provided the technical support and guidance. All authors have read and agreed to the published version of the manuscript.

**Funding:** Higher Education Commission (HEC) Pakistan provided funding through project# 20-3844/R&D/HEC/14.

**Institutional Review Board Statement:** All authors confirm that they followed all ethical guidelines. All authors certify that they have no affiliations with or involvement in any organization or entity with any financial interest or non-financial interest in the subject matter or materials discussed in this manuscript.

**Informed Consent Statement:** Not applicable.

**Data Availability Statement:** Data is contained within the article.

**Acknowledgments:** Authors are thankful to the National Research Program for Universities (NRPU), Higher Education Commission (HEC) Pakistan for provision of funding through project# 20-3844/R&D/HEC/14. Authors are also thankful to the Mechanical Engineering Department, Institute of Space Technology for the Charpy test facility and Failure Analysis Center, IST for stereomicrographic analysis of the composites. Authors are thankful to the School of Chemical and Materials Engineering, NUST, for tensile testing of the NiTi wire.

**Conflicts of Interest:** The authors declare that they have no conflict of interest.

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
