# Peer review of "Impact Energy Absorption Analysis of Shape Memory Hybrid Composites"

_jcs, doi:10.3390/jcs6120365_

Round 1

Reviewer 1 Report

The topic of investigation in this study is the investigation of mechanical and impact resistance of polymer matrix composites additionally reinforced with shape memory alloy wires. The topic is interesting, however, highlighting the goals are necessary (see detailed comments). In numerous cases, there are lacks of important information, which need to be added or extended according to the comments below.

  1. Line 71 and further: please check the correctness of symbols, moreover, variables should be written using italic font.
  2. It is recommended to highlight the research goal in this study, underlining what original input was performed by the authors compared to previous activities of the team and activities of other research group working in this thematic area.
  3. Please provide necessary details on constituents (manufacturers, trade names of the materials, etc.) as well as details on used equipment during manufacturing (models of devices, manufacturers) to make the manufacturing process reproducible.
  4. Please provide information of the microscope used together with settings and parameters of observation.
  5. Please provide information on impact testing device, and provide and justify impact energy values.
  6. The presence of metallic wires in a polymer matrix composites creates stress concentrations during mechanical loading due to significant difference in properties. It is thus essential to address to structural changes during impact, especially fracture mechanisms occurred on the interfaces. The extended discussion on the observed mechanisms with addressing to fracture mechanics concepts for justification need to be provided.
  7. The discussion on the effect of notch presence on impact resistance provided in the first paragraph of section 3.2 need to be supported with explanation of mechanical behavior of the material. Moreover, addressing to BVID in the next paragraph it is likely to compare the obtained results to the classical GFRP composites described in the literature and highlight the differences of impact response of the investigated material compared to classical GFRPs.
  8. In section 3.3, a part regarding treatment of wires is a repetition with respect to the description provided in section 2, please make sufficient corrections.
  9. Please discuss how the deformation of SMA wires affect structural integrity of the GFRP with SMA reinforcement.
  10. It is recommended to enrich conclusions with quantitative results obtained in the study.
  11. The authors should format the manuscript according to the requirements of the journal. Minor grammar and punctuation errors need to be corrected.

Author Response

Reviewer #1:

Comments:

  1. Line 71 and further: please check the correctness of symbols, moreover, variables should be written using italic font.

Our Answer: The font of symbols and variables is corrected and written in italic and are highlighted.

  1. It is recommended to highlight the research goal in this study, underlining what original input was performed by the authors compared to previous activities of the team and activities of other research group working in this thematic area.

Our Answer: The goals of the research work are described in lines 111-121 with reference to literature of previous research.

  1. Please provide necessary details on constituents (manufacturers, trade names of the materials, etc.) as well as details on used equipment during manufacturing (models of devices, manufacturers) to make the manufacturing process reproducible.

Our Answer: The manufacturers of the wires is mentioned in section 2 and highlighted in track change mode. The glass fibers used are E-class glass fibers. The vacuum pump used is also mentioned in section 2.

  1. Please provide information of the microscope used together with settings and parameters of observation.

Our Answer: The microscopes used are optical and stereo microscopes. Their model and magnification are mentioned in section 3 (lines 205-206), and section 3.1.1 lines (314-317).

  1. Please provide information on impact testing device, and provide and justify impact energy values.

Our Answer: The impact testing device information is provided in section 2 (lines 197-198) and impact energy values are tabulated in Tables 1 and 2. These are the average values of toughness and impact strength of the manufactured composites with minor fluctuations as indicated by the error bars in graphs of Figures 8 and 9.

  1. The presence of metallic wires in a polymer matrix composites creates stress concentrations during mechanical loading due to significant difference in properties. It is thus essential to address to structural changes during impact, especially fracture mechanisms occurred on the interfaces. The extended discussion on the observed mechanisms with addressing to fracture mechanics concepts for justification need to be provided.

Our Answer: It is true that metallic wires in polymer composites create stress concentration. The interfacial strength of metals with matrix is improved by increasing surface area of interface by improving surface roughness (treatment mentioned in lines 156-160), that creates the mechanical interlocking between wires and matrix. Due to ductile behavior of wires, they are found to improve the overall toughness of the hybrid composites on impact loading. The fracture mechanisms are further discussed in section 3.1.3 and 3.2 providing further justifications. Future work in this area is in progress.

  1. The discussion on the effect of notch presence on impact resistance provided in the first paragraph of section 3.2 need to be supported with explanation of mechanical behavior of the material. Moreover, addressing to BVID in the next paragraph it is likely to compare the obtained results to the classical GFRP composites described in the literature and highlight the differences of impact response of the investigated material compared to classical GFRPs.

Our Answer: The discussion on effect of notch has been made in accordance with reviewer’s comment. The extended discussed has been done in accordance with literature survey in section 3.1.2 (lines 357-367) and in the last paragraph of 3.1.2. BVIDs are addressed to give the likelihood of resolving this issue during processing.

  1. In section 3.3, a part regarding treatment of wires is a repetition with respect to the description provided in section 2, please make sufficient corrections.

Our Answer: The relevant correction has been made and repetition is erased.

  1. Please discuss how the deformation of SMA wires affect structural integrity of the GFRP with SMA reinforcement.

Our Answer: The improvement in structural integrity has been discussed in more detail, as suggested by the reviewer, in section 3.1.3 (highlighted in track mode) and 3.2 with respect to fractography.

  1. It is recommended to enrich conclusions with quantitative results obtained in the study.

Our Answer: Qualitative values are mentioned in conclusions as suggested by reviewer.

  1. The authors should format the manuscript according to the requirements of the journal. Minor grammar and punctuation errors need to be corrected.

Our Answer: The references are edited as per journal instructions and minor grammar and punctuation changes are done, highlighted in track mode.

Reviewer 2 Report

This manuscript is very novel and interesting.

However, some issues need to be resolved.

1. There are too many conclusions that need to be shortened.

2. Please explain the application value of this research.

3. Please add in the summary why you want to improve the impact.

4. Please add statistical differences to the pictures.

Author Response

Reviewer 2

Comments and Suggestions for Authors

  1. There are too many conclusions that need to be shortened.

Our Answer: The conclusions 2, 6 and 8 have been erased.

  1. Please explain the application value of this research.

Our Answer: The application areas of this research are mentioned in section 1 (lines 126-130)

  1. Please add in the summary why you want to improve the impact.

Our Answer: The goals of the research work are described in lines 111-121, with application industries mentioned in section 1 (lines 126-130).

  1. Please add statistical differences to the pictures.

Our Answer: Statistical analysis of the pictures (Figures 8, 9 and 17), has been done using ANOVA, and the description is provided in captions and text, and highlighted in track change mode.

Round 2

Reviewer 1 Report

The authors introduced significant improvements in the manuscript according to the comments from the previous review report. Additionally, they provided comprehensive answers to the comments with references to the changes made in the manuscript. According to this, I recommend this manuscript for a publication in its present from.

Reviewer 2 Report

The manuscript has been revised.

However, Figure 8 is still uncorrected. thanks.

The rest I think is acceptable.